# The Clinical Outcomes Based on the Achievement of Low-Density Lipoprotein Cholesterol Targets after ST Elevation Myocardial Infarction

**DOI:** 10.3390/jcm9010079

**Published:** 2019-12-28

**Authors:** Wei-Chieh Lee, Chih-Yuan Fang, Chien-Jen Chen, Cheng-Hsu Yang, Chiung-Jen Wu, Hsiu-Yu Fang

**Affiliations:** 1Division of Cardiology, Department of Internal Medicine, Kaohsiung Chang Gung Memorial Hospital, Chang Gung University College of Medicine, Kaohsiung 833, Taiwan; leeweichieh@yahoo.com.tw (W.-C.L.); cyfang@seed.net.tw (C.-Y.F.); cjchen@cloud.cgmh.org.tw (C.-J.C.); yangch@cloud.cgmh.org.tw (C.-H.Y.); cvcjwu@cloud.cgmh.org.tw (C.-J.W.); 2Institute of Clinical Medicine, College of Medicine, National Cheng Kung University, Tainan 701, Taiwan

**Keywords:** ST elevation myocardial infarction, low-density lipoprotein cholesterol, clinical outcomes, left main coronary artery disease

## Abstract

The clinical outcome of patients with ST elevation myocardial infarction (STEMI) undergoing primary percutaneous coronary intervention (PCI), with or without achievement of low-density lipoprotein cholesterol (LDL-C) targets, has rarely been investigated. This study was performed to investigate the comparison of clinical outcome in STEMI patients with or without achievement LDL-C targets (below 70 mg/dL and/or ≥50% reduction). Between November 2013 and December 2016, 689 STEMI patients underwent primary PCI in our hospital. Patients who were deceased, lost to follow-up, had no follow-up lipid profile, or had no side effects after statin use were excluded. A total of 343 patients were classified into group 1 (with LDL-C target achievement) and 172 patients were classified into group 2 (without LDL-C target achievement). Between the two groups, a higher prevalence of left main coronary artery disease, smaller pre-PCI stenosis, and a larger pre-PCI minimal luminal diameter were noted in group 2. The incidence of post-MI angina (8.7% vs. 6.4%; *p* = 0.393), target vessel revascularization (2.3% vs. 3.5%; *p* = 0.566), and recurrent MI (1.5% vs. 1.2%; *p* = 1.000), showed similar results between the two groups during a one-year follow-up period. Initial LDL-C levels ≥130 mg/dL, left main coronary artery disease, and absence of diabetes mellitus were positively associated with non-achievement of LDL-C targets. After STEMI, 66.6% of patients could achieve LDL-C targets one year later. However, such patients did not show better clinical outcomes. Non-DM, initial LDL-C levels ≥130 mg/dL, and left main coronary artery disease were related to non-achievement of LDL-C targets.

## 1. Introduction

Hyperlipidemia is a major risk factor for coronary heart disease, and it is well known that treatment of hyperlipidemia reduces the morbidity and mortality of coronary heart disease [1]. In both Asian and non-Asian populations, the risk of acute myocardial infarction (MI) is associated with an increase in low-density lipoprotein cholesterol (LDL-C) and a decrease in high-density lipoprotein cholesterol (HDL-C) [2]. In addition, molecular and cellular studies have established a central role for LDL-C in the pathogenesis of atherosclerotic plaques, and their clinical sequelae, including coronary heart disease and ischemic stroke [3]. Therefore, lowering LDL-C and achieving LDL-C target with a statin is important in both primary and secondary intervention settings [4]. Current European Society of Cardiology/European Atherosclerosis Society joint guidelines emphasize that LDL-C is still the most important marker to treat targets, regardless of LDL-C levels, with a target LDL-C <70 mg/dL or a reduction of ≥50% if the baseline is between 70 and 135 mg/dL. Statin therapy should be initiated for very high-risk populations [5]. In a large meta-analysis study, patients who achieved very low LDL-C levels (<50 mg/dL or 50 to <70 mg/dL) had a lower risk for major cardiovascular disease events than those who achieved moderately low levels (75 to <100 mg/dL) [6].

In real-world practice, most post-MI patients with hyperlipidemia do not attain LDL-C targets, especially ST elevation MI (STEMI) and renal insufficiency patients [7,8]. On the other hand, most post-MI patients receive low- or moderate-intensity statins and around 66% of patients are estimated to be discharged on a statin and reach guideline-recommended LDL-C targets [9]. Large interindividual variations in response to statin therapy have also been noted [6]. Otherwise, there are still arguments for the use of statins in low LDL-C patients, and there has not been a full evaluation on the impact of the intensity of statin therapy, as represented by the achieved level of LDL-C, on cardiovascular outcomes in patients with acute MI [10].

Because of this gap in knowledge, the present study aimed to explore the clinical outcome in STEMI patients with or without achievement of LDL-C targets (below 70 mg/dL and/or ≥50% reduction) one year after STEMI.

## 2. Materials and Methods

### 2.1. Patients and Groups

Between November 2013 and December 2016, 689 STEMI patients underwent primary percutaneous coronary intervention (PCI) in our hospital. A total of 126 patients without statin use had no follow-up lipid profile, 38 patients expired within a one-year follow-up period, and 10 patients had severe liver disease, all of which were excluded. A total of 515 patients were recruited in this study (Figure 1). A total of 343 patients were classified into group 1 (with LDL-C target achievement) and 172 patients were classified into group 2 (without LDL-C target achievement).

Samples for baseline laboratory tests, except for lipid measurements, were collected at admission or before primary PCI. Then, the patients were required to fast at least 12 h before lipid profile, HbA1C, and fasting sugar were measured. LDL-C levels were calculated by the spectrophotometric method. The baseline characteristics, angiographic characteristics, one-year clinical outcomes, and changes of follow-up lipid profiles were compared among the two groups. The Institutional Review Committee on Human Research at our institution approved the study protocol.

All procedures followed were in accordance with the ethical standards of the responsible committee on human experimentation (institutional and national) and with the Helsinki Declaration of 1964 and later revisions.

### 2.2. Follow-Up

In a one-year follow-up period, the lipid profile was measured every 3–6 months. All patients received moderate-intensity statins (atorvastatin 20 mg daily or rosuvastatin 10 mg daily) first, and dosage was modified based on the side effects and follow-up lipid profile.

### 2.3. Definitions

The LDL-C target was defined as LDL-C levels below 70 mg/dL and/or ≥50% reduction at a one-year follow-up period. Our MI criteria were in accordance with the universal definitions [11]. Advanced heart failure was graded as greater than III, according to the New York Heart Association Classification. Post-MI angina was defined as the patients experiencing typical angina which was characterized by at least one of the following symptoms: chest pain occurring at rest or during minimal exertion and usually lasting less than 20 min (without nitroglycerin administration), new onset of severe flank pain, or a crescendo pattern of chest pain (more severe, prolonged, or at an increased frequency than previously experienced) [12]. Based on the Kidney Disease Improving Global Outcomes definition, acute kidney injury (AKI) was defined as an absolute increase in serum creatinine of at least 0.3 mg/dL within 48 h or a 50% increase in serum creatinine from baseline within 7 days, or a urine volume of less than 0.5 mL/kg/h for at least 6 h [13]. Target vessel revascularization (TVR) was defined as any repeat PCI or coronary artery bypass graft for lesions with stenosis ≥70%, and the target vessel was defined as the entire major coronary vessel proximal and distal to the target lesion, including upstream and downstream branches and the target lesion itself [14].

### 2.4. Study Endpoints

The primary endpoints were recurrent angina episode that needed admission or an emergency department visit, recurrent TVR, recurrent MI, and stroke during the one-year follow-up period. The secondary endpoints of our study were LDL-C levels <70 mg/dL and or ≥50% reduction compared to initial LDL-C level.

### 2.5. Statistical Analysis

Data were expressed as the mean ± standard deviation for continuous variables, or as counts and percentages for categorical variables. Continuous variables were compared using an independent *t*-test. Categorical variables were compared using a chi-square statistic. Univariate logistic regression analysis was performed to determine the odds ratios (ORs) for non-achievement of LDL-C targets. A significant association in the univariate logistic regression analysis (left main coronary artery disease) and variables of univariate logistic regression analyses with a *p*-value ≤ 0.200 were included in the multivariate logistic regression analysis to determine the associations of LDL-C target non-achievement. All statistical analyses were performed by using SPSS 22.0 (IBM. Corp., Armonk. NY, USA). A *p*-value < 0.05 was defined as statistically significant.

## 3. Results

### 3.1. Baseline Characteristics of Study Groups

Baseline characteristics of the two groups are listed in Table 1. The average age of group 1 was 60 ± 12.7 years, and the average age of group 2 was 59 ± 12.9 years (*p* = 0.211). The predominant gender was male in both groups. The prevalence of comorbidities and the severities of MI showed no significant differences between the two groups. Laboratory data were similar between the two groups, including the average LDL-C level (group 1 vs. group 2; 109.32 ± 40.58 mg/dL vs. 113.49 ± 34.16 mg/dL; *p* = 0.247), the prevalence of LDL-C ≥130mg/dL (group 1 vs. group 2; 24.2% vs. 32.0%; *p* = 0.073), the average HDL-C level (group 1 vs. group 2; 41.11 ± 10.75 mg/dL vs. 41.04 ± 12.33 mg/dL; *p* = 0.943), and the prevalence of HDL-C (group 1 vs. group 2; 50.4% vs. 46.5%; *p* = 0.162). Left ventricular ejection fraction, infarcted territory, the prevalence of multivessel coronary artery disease, and post-MI medication use did not show a significant difference between the two groups. In group 2, a higher prevalence of left main coronary artery disease was noted when compared to group 2 (group 1 vs. group 2; 4.7% vs. 9.9%; *p* = 0.034). The percentage of increasement of the statin dose (group 1 vs. group 2; 16.0% vs. 30.8%; *p* < 0.001) and ezetimibe use (group 1 vs. group 2; 2.6% vs. 12.8%; *p* < 0.001) was significant higher in group 2.

### 3.2. Angiographic Characteristics of Study Groups

Angiographic characteristics of the two groups are listed in Table 2. Higher severities of pre-PCI stenosis (group 1 vs. group 2; 95.15 ± 8.35% vs. 93.46 ± 9.82%; *p* = 0.046) and smaller pre-PCI minimal luminal diameter (MLD) (group 1 vs. group 2; 0.14 ± 0.05 mm vs. 0.23 ± 0.12 mm; *p* = 0.005) were noted in group 1. Pre-PCI reference luminal diameter (RLD), post-PCI stenosis, post-PCI MLD, and post-PCI RLD showed no significant difference between the two groups. The method of reperfusion and the mechanical supportive devices showed similarities between the two groups. The incidence of post-PCI acute kidney injury was similar between the two groups (group 1 vs. group 2; 8.7% vs. 6.4%; *p* = 0.393).

### 3.3. One-Year Clinical Outcomes of Study Patients

One-year clinical outcomes were listed in Table 3. During the one-year follow-up period, the incidence of recurrent post-MI angina (group 1 vs. group 2; 8.7% vs. 9.3%; *p* = 0.870), target vessel revascularization (group 1 vs. group 2; 2.3% vs. 3.5%; *p* = 0.556), and recurrent MI (group 1 vs. group 2; 1.5% vs. 1.2%; *p* = 1.000), and stroke (group 1 vs. group 2; 0.3% vs. 0%; *p* = 1.000) showed similar results between the two groups.

### 3.4. The Change in LDL-C Levels of the Study Groups at One-Year Follow-up

At one-year follow-up, the patients in group 1 had higher average of LDL-C levels when compared to group 2 (group 1 vs. group 2; 54.24 ± 15.18 mg/dL vs. 92.73 ± 22.48 mg/dL; *p* < 0.001) (Table 4). In group 1, 94.2% of patients achieved an LDL-C level ≤ 70 mg/dL, and 50.1% of patients achieved a 50% reduction in LDL-C level. There was a significant difference between the two groups when comparing the average reduction in LDL-C level (group 1 vs. group 2; −55.08 ± 36.86 mg/dL vs. −20.77 ± 36.78 mg/dL; *p* < 0.001) and the average percentage of LDL-C reduction (group 1 vs. group 2; −44.61 ± 25.12% vs. −6.79 ± 32.86%; *p* < 0.001).

### 3.5. The Serial Changes of Lipid Profile in the Two Groups

In group 1, the average LDL-C level decreased to 72.39 ± 25.36 mg/dL at 6-month follow-up period and showed significant difference when comparing with initial LDL-C level (Figure 2). At one-year follow-up period, the average LDL-C level decreased to 54.24 ± 15.18 mg/dL and presented significant difference when comparing with 6-month LDL-C level. In group 2, the average LDL-C level decreased to 104.79 ± 28.20 mg/dL at 6-month follow-up period and showed significant difference when comparing with initial LDL-C level (Figure 2). In the one-year follow-up period, the average LDL-C level decreased to 92.73 ± 22.48 mg/dL and presented significant difference when comparing with the 6-month LDL-C level.

### 3.6. Univariate and Multivariate Logistic Regression Analyses of Associations for Non-Achievement of LDL-C Targets

Univariate logistic regression analyses for associations for non-achievement of LDL-C targets in all study patients showed a positive association with left main coronary artery disease (OR: 2.242; 95% confidence interval (CI): 1.103–4.554; *p* = 0.026) (Table 5). Multivariate logistic regression analyses (Table 5) were performed and found a positive association of left main coronary artery disease, with the p-value of the multivariate logistic regression analysis ≤0.200. Diabetes mellitus (DM) (OR: 0.651; 95% CI: 0.435–0.991; *p* = 0.045) had a negative association. Initial LDL-C levels ≥130 mg/dL (OR: 1.583; 95% CI: 1.038–2.415; *p* = 0.033) and left main coronary artery disease (OR: 2.376; 95% CI: 1.149–4.915; *p* = 0.020) were positively associated with non-achievement of LDL-C targets.

### 3.7. The Achieved LDL-C Target Rate and the Incidence of Re-MI, TVR, and Stroke in Different Subgroups

The comparison of subgroups between presence and absence of left main coronary artery disease, multiple vessel coronary artery disease, and single-vessel coronary artery disease follow-up LDL-C levels ≥100 mg/dL and <100 mg/dL are presented in Figure 3. A total of 51.5% (17/33) patients with left main coronary artery disease and 32.2% (155/482) patients without left main coronary artery disease did not achieve LDL-C targets (*p* = 0.034). The incidence of one-year re-MI, TVR, and stroke did not show a significant difference between these two subgroups. A total of 32.9% (105/319) patients with multiple vessel coronary artery disease and 34.2% (67/196) patients with single-vessel coronary artery disease did not achieve LDL-C targets (*p* = 0.774). The incidence of one-year re-MI, TVR, and stroke was not significantly different between these two subgroups. A total of 92.9% (39/42) patients with follow-up LDL-C levels ≥100 mg/dL and 28.1% (133/473) patients with follow-up LDL-C levels <100 mg/dL did not achieve LDL-C targets (*p* < 0.001). The incidence of one-year re-MI, TVR, and stroke was 7.7% (3/39) in the patients with follow-up LDL-C levels ≥100 mg/dL and showed no significant difference when compared to the patients with follow-up LDL-C levels <100 mg/dL.

## 4. Discussion

Three important findings of our study are made known. First, 66.6% (343/515) of STEMI patients achieved an LDL-C target with a moderate-intensity statin at a one-year follow-up period. Second, clinical outcomes were similar including post-MI angina, TVR, and recurrent MI, between LDL-C target achievers and LDL-C target non-achievers. Non-DM, an initial LDL-C level ≥130 mg/dL, and left main coronary artery disease were positively associated with non-achievement of LDL-C targets. Third, dyslipidemia and prior statin use was not a predictor about the achievement of LDL-C target.

Current guidelines recommend statin therapy should be initiated after acute MI and recommend an LDL-C target of <100 mg/dL for patients with established coronary heart disease and a target of <70 mg/dL for acute coronary syndrome patients [5]. Statins are considered a first-line therapy to achieve such reductions in LDL-C levels, but most acute coronary syndrome patients fail to achieve their LDL-C target of <70 mg/dL [15]. The most common reasons were infrequent use of high-intensity statins, the absence of a statin prescription at discharge, lack of persistence on statin therapy, nonwhite race, and lack of insurance [9,16]. In the recent study about proprotein convertase subtilisin–kexin type 9 (PCSK9) inhibitor use for secondary prevention after acute coronary syndrome, the patients with lower LDL-C level had better outcomes about recurrent ischemic cardiovascular events during the longer follow-up period [17].

In Asian patients, lower statin doses achieve lipid improvements when compared with those observed with higher doses in Caucasians because of genetic differences in the metabolism of statins, hepatic enzymes, and drug transporters [18]. On the other hand, myopathy after statin use happens frequently in the Asian population, smaller body size, female gender, and old age groups [19]. In real-world practice, some patients can achieve an LDL-C target through lifestyle modification [20]. Therefore, we may focus on results after the achievement of LDL-C targets and not just focus on the use of high-intensity statins in the Asian population.

Achievement of LDL-C targets in patients with cardiovascular disease has not been fully evaluated, especially in the Asian population. Natsuaki et al. reported the risks for cardiovascular events were comparable if an LDL-C level <120 mg/dL was achieved [10]. The risk for MACEs was significantly higher in the LDL-C ≥120 mg/dL group than in patients with LDL-C levels between 80–99 mg/dL; however, the risk for MACEs was not significantly lower in the LDL-C <80 mg/dL group [10]. T Ahn et al. reported acute MI patients that achieved LDL-C targets did not show better clinical outcomes [21]. Lee et al. also found a significantly better survival of MI patients if patients attained LDL-C <2.4 mmol/L; however, this difference became insignificant when an LDL-C level of <2.3 mmol/L was taken [22]. According to the above studies, the clinical benefits associated with the LDL-C target <70 mg/dL in Asian patients remains controversial. In our study, we focused on STEMI patients and did not observe clinical benefits if LDL-C targets were achieved. In the subgroups with follow-up LDL-C levels ≥100 mg/dL, the incidence of one-year MACE was 7.7% (3/39) but this was not significantly different when compared to the patients with follow-up LDL-C levels <100 mg/dL.

In our study, 42.5% (219/515) of STEMI patients had prior dyslipidemia under statin therapy. In such patients, non-DM, an initial LDL-C level ≥ 130 mg/dL, and left main coronary artery disease were related to non-achievement of LDL-C targets. Physicians may give more aggressive lipid control and sugar control in the DM patients. Therefore, non-DM became the predictor of non-achievement of LDL-C target. In our study, the prevalence of initial LDL-C level ≥130 mg/dL was also higher in non-DM patients (non-DM vs. DM; 96/334 (28.7%) vs. 42/181 (23.2%); *p* = 0.211). In the patients with left main coronary artery disease, a higher prevalence of LDL-C target non-achievers was noted at one-year follow-up when compared with the patients without left main coronary artery disease. In one prospective observational study, combined hyperlipidemia, simple hypercholesterolemia, metabolic syndrome, and low HDL-C levels were associated with multiple vessel coronary artery disease and the patients with combined hyperlipidemia presented the highest prevalence of left main coronary artery disease [23]. For the patients with left main coronary artery disease, aggressive lipid control contributes less to atherosclerosis progression [23]. On the other hand, the patients with left main coronary artery disease had the possibilities about non-achievement of LDL-C target even though aggressive lipid-lower agents. Therefore, more aggressive antihyperlipidemic agents need to be considered for STEMI patients with left main coronary artery disease to achieve LDL-C targets. In one observational study about patients with the acute coronary syndrome, a higher prevalence of left main coronary artery disease was noted in patients who did not attain LDL target, even though the p-value did not reach a significant difference [24]. In our study, the prevalence of initial LDL-C level ≥130 mg/dL was also higher in patients with left main coronary artery disease (non-LM vs. LM; 128/482 (26.6%) vs. 10/33 (30.3%); *p* = 0.685). On the other hand, an initial LDL-C level ≥130 mg/dL was the major problem with the non-achievement of LDL-C targets.

As a limitation, this was a retrospective study from a single medical center. We only focused on STEMI patients in the Asian population and used moderate-intensity statins first. This strategy reflected the real-world practice in Asian countries because high-intensity statins may bring more side effects on small-sized Asians. The current strategy for lipid control has only focused on lowering the serum LDL-C level no matter which ethnicity. Another limitation was no PCSK-9 inhibitor was used during study period in our study. Our study demonstrated that STEMI patients did not have better outcomes after achieving LDL-C targets. Left main coronary artery disease was shown to be related to non-achievement of LDL-C targets. More aggressive lipid control strategies need to be considered in STEMI patients with left main coronary artery disease. We still need more research about lipid control in the STEMI patients with non-DM and left main coronary artery disease.

## 5. Conclusions

One year after STEMI, 66.6% of patients achieved LDL-C targets. However, such patients did not show better clinical outcomes. Non-DM, an initial LDL-C level ≥130 mg/dL, and left main coronary artery disease were related to non-achievement of LDL-C targets.

## Figures and Tables

**Figure 1 jcm-09-00079-f001:**
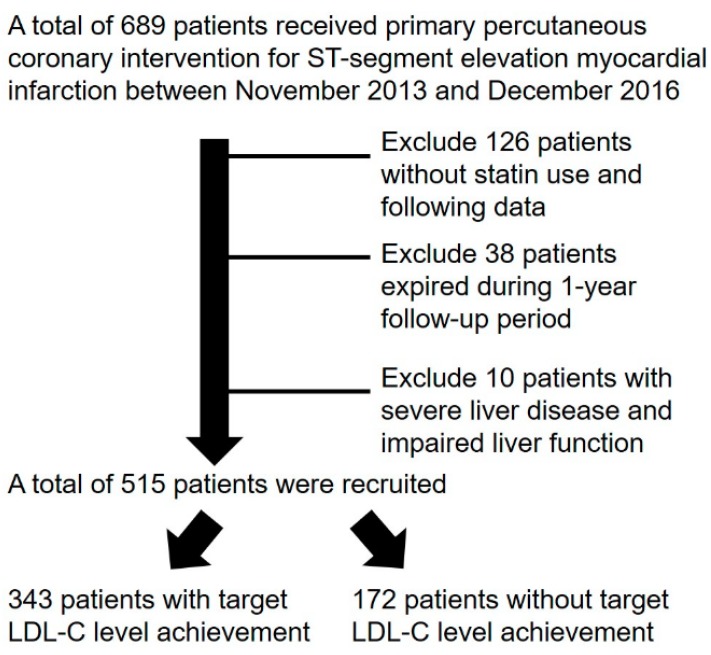
Patients disposition. Abbreviation: LDL-C: low-density lipoprotein cholesterol.

**Figure 2 jcm-09-00079-f002:**
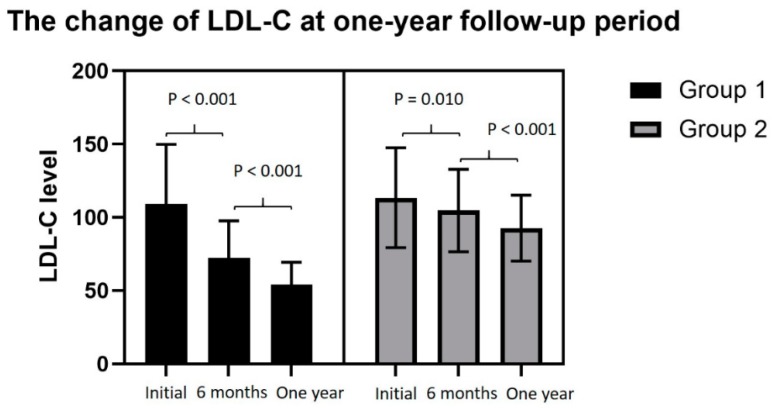
In the two groups, the serial changes of lipid profile decreased gradually. There was significant difference when initial LDL-C level was compared with 6-month LDL-C level and the 6-month LDL-C level compared with one-year LDL-C level.

**Figure 3 jcm-09-00079-f003:**
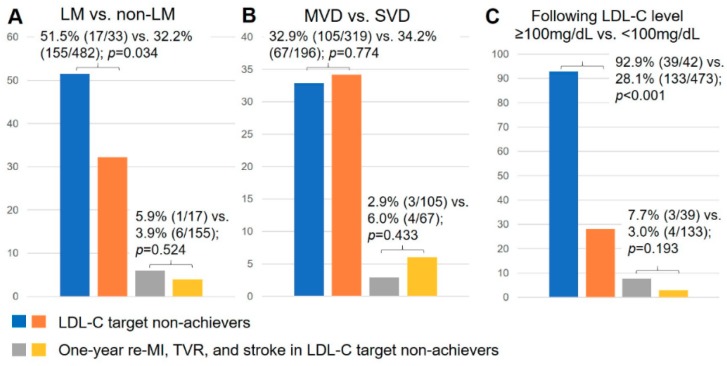
The comparison of subgroups between presence and absence of left main (LM) coronary artery disease, multiple vessel disease (MVD) and single vessel disease (SVD), following LDL-C level ≥100 mg/dL and <100 mg/dL. (**A**): The comparison between the patients with or without LM coronary artery disease: 51.5% (17/33) patients with LM coronary artery disease and 32.2% (155/482) patients without LM coronary artery disease did not achieve LDL-C target (*p* = 0.034). The incidence of a one-year re-MI, TVR, and stroke did not have a significant difference between these two subgroups (LM vs. non-LM; 5.9% (1/17) vs. 3.9% (6/155); *p* = 0.524). (**B**): The comparison between the patients with MVD and SVD: 32.9% (105/319) patients with MVD and 34.2% (67/196) patients with SVD did not achieve LDL-C target (*p* = 0.774). The incidence of one-year re-MI, TVR, and stroke was not significantly different between these two subgroups (MVD vs. SVD; 2.9% (3/105) vs. 6.0% (4/67); *p* = 0.433). (**C**): The comparison between the patients with following LDL-C level ≥100 mg/dL and <100 mg/dL: 92.9% (39/42) patients with following LDL-C level ≥100 mg/dL and 28.1% (133/473) patients with following LDL-C level <100 mg/dL did not achieve LDL-C target (*p* < 0.001). The incidence of one-year re-MI, TVR, and stroke showed no significant difference between these two subgroups (following LDL-C level ≥100 mg/dL vs. <100 mg/dL; 7.7% (3/39) vs. 3.0% (4/133); *p* = 0.193).

**Table 1 jcm-09-00079-t001:** Baseline characteristics of study patients.

	Group 1 (*N =* 343)	Group 2 (*N =* 172)	*p*-Value
General demographics			
Age (years)	60 ± 12.7	59 ± 12.9	0.211
Male sex (%)	284 (82.8)	135 (78.5)	0.609
BMI (kg/m^2^)	26.20 ± 9.46	25.87 ± 4.04	0.668
Comorbidities			
Diabetes mellitus (%)	128 (37.3)	53 (30.8)	0.171
Current smoker (%)	204 (59.5)	105 (61.0)	0.775
Hypertension (%)	199 (58.0)	110 (64.0)	0.215
Prior MI (%)	13 (3.8)	9 (5.2)	0.491
Prior stroke (%)	17 (5.0)	6 (3.5)	0.506
CKD stage ≥ 3 (eGFR < 60 mL/min/1.73 m^2^)	45 (13.1)	20 (11.6)	0.675
ESRD on maintenance hemodialysis (%)	9 (2.6)	3 (1.7)	0.759
Dyslipidemia with prior statin use (%)	144 (42.0)	75 (43.6)	0.777
Advanced heart failure (%)	3 (0.9)	1 (0.6)	1.000
The severity of MI			
SBP (mmHg)	139.82 ± 33.28	141.67 ± 32.90	0.551
Killip level (%)			0.721
I, II	295 (86.0)	146 (84.9)	
III, IV	48 (14.0)	26 (15.1)	
Timing of primary PCI			
Door-to-balloon time (min)	69.04 ± 45.04	83.64 ± 46.17	0.094
Reperfusion time (min)	17.81 ± 7.39	18.82 ± 8.42	0.167
Pain-to-reperfusion time (min)	225.49 ± 156.17	208.59 ± 101.83	0.575
Laboratory examination			
White blood cell count (×10^3^)	11.1 ± 3.6	11.0 ± 3.7	0.788
Hemoglobin (gm/dL)	14.5 ± 2.0	14.6 ± 2.0	0.595
Blood fasting sugar (mg/dL)	180.8 ± 90.5	173.6 ± 82.6	0.421
HbA1C (%)	6.92 ± 1.85	6.71 ± 1.56	0.200
Creatinine (except ESRD) (mg/dL)	1.11 ± 0.51	1.08 ± 0.36	0.475
Total cholesterol (mg/dL)	178.96 ± 49.25	185.11 ± 42.27	0.164
LDL-cholesterol (mg/dL)	109.32 ± 40.58	113.49 ± 34.16	0.247
>130 mg/dL (%)	83 (24.2)	55 (32.0)	0.073
HDL-cholesterol (mg/dL)	41.11 ± 10.75	41.04 ± 12.33	0.943
<40 mg/dL (%)	173 (50.4)	80 (46.5)	0.455
Peak troponin-I (ng/mL)	50.93 ± 35.99	46.25 ± 34.12	0.162
LVEF (%)	57.40 ± 12.71	56.32 ± 12.60	0.366
Infarcted territory (%)			0.400
Anterior wall	184 (53.6)	85 (49.4)	
Non-anterior wall	159 (46.4)	87 (50.6)	
Characteristics of coronary artery disease			
Single or multiple vessel disease (%)			0.983
Single vessel disease	129 (37.6)	67 (39.0)	
Multiple vessel disease	214 (62.4)	105 (61.0)	
Non-culprit lesion stenosis ≥ 70% (%)	144 (67.3)	76 (72.4)	0.210
Left main coronary artery disease (%)	16 (4.7)	17 (9.9)	0.034
Post-MI Medications			
ACEI/ARBs (%)	314 (94.3)	153 (91.1)	0.190
Beta-blockers (%)	298 (89.5)	148 (88.1)	0.328
Kept the same brand of statin therapy	316 (92.1)	151 (87.8)	0.147
The change of lipid-lower strategy			
Increasement of the statin dose (%)	55 (16.0)	53 (30.8)	< 0.001
Add ezetimibe (%)	9 (2.6)	22 (12.8)	< 0.001

Data are expressed as mean ± standard deviation or as number (percentage). Abbreviation: MI: myocardial infarction; CKD: chronic kidney disease; BMI: body mass index; ESRD: end stage renal disease; SBP: systolic blood pressure; HbA1C: glycohemoglobin; LDL: low density lipoprotein; HDL: high density lipoprotein; LVEF: left ventricular ejection fraction; ACEI: angiotensin converting enzyme inhibitor; ARB: angiotensin receptor blocker.

**Table 2 jcm-09-00079-t002:** Angiographic characteristics of study patients.

	Group 1 (*N =* 343)	Group 2 (*N =* 172)	*p*-Value
Primary PCI angiography			
Culprit vessel			
Pre-PCI TIMI flow			0.192
≥2 (%)	79 (23.6)	49 (29.3)	
≤1 (%)	264 (76.4)	123 (70.7)	
Pre-PCI stenosis (%)	95.15 ± 8.35	93.46 ± 9.82	0.046
Pre-PCI MLD (mm)	0.14 ± 0.05	0.23 ± 0.12	0.005
Pre-PCI RLD (mm)	3.20 ± 0.57	3.25 ± 0.60	0.380
Post-PCI TIMI flow			0.668
≥2 (%)	340 (99.1)	170 (98.8)	
≤1 (%)	3 (0.9)	2 (1.2)	
Post-PCI stenosis (%)	12.62 ± 10.31	12.46 ± 7.23	0.856
Post-PCI MLD (mm)	2.86 ± 0.56	2.84 ± 0.50	0.708
Post-PCI RLD (mm)	3.34 ± 0.52	3.35 ± 0.58	0.856
Distal embolization (%)	12 (3.5)	6 (3.5)	1.000
Method of perfusion			0.442
Balloon angioplasty alone (%)	15 (4.4)	12 (7.0)	
Bare-metal stents (%)	123 (35.9)	58 (33.7)	
Drug-eluting stents (%)	205 (59.8)	102 (59.3)	
Procedural device			
IABP (%)	49 (14.3)	22 (12.8)	0.686
ECMO (%)	4 (1.2)	2 (1.2)	1.000
Post PCI acute kidney injury (%)	30 (8.7)	11 (6.4)	0.393

Data are expressed as mean ± standard deviation or as number (percentage). Abbreviation: DM: diabetes mellitus; TG: Triglyceride; HDL-C: high-density lipoprotein cholesterol; PCI: percutaneous coronary intervention; TIMI: thrombolysis in myocardial infarction; MLD: minimal luminal diameter; RLD: reference luminal diameter; CABG: coronary artery bypass graft; IABP: intra-aortic balloon pumping; ECMO: extracorporeal membrane oxygenation.

**Table 3 jcm-09-00079-t003:** One-year clinical outcomes of study patients.

	Group 1 (*N =* 343)	Group 2 (*N =* 172)	*p*-Value
Recurrent angina (%)	30 (8.7)	16 (9.3)	0.870
Target-vessel revascularization (%)	8 (2.3)	6 (3.5)	0.566
Recurrent myocardial infarction (%)	5 (1.5)	2 (1.2)	1.000
Stroke (%)	1 (0.3)	0 (0)	1.000

Data are expressed as numbers (percentage).

**Table 4 jcm-09-00079-t004:** The change in LDL-C levels of the study groups at one-year follow-up.

	Group 1 (*N =* 343)	Group 2 (*N =* 172)	*p*-Value
LDL-C level (mg/dL)	54.24 ± 15.18	92.73 ± 22.48	< 0.001
LDL-C < 70 mg/dL (%)	323 (94.2)	0 (0)	< 0.001
The change of reduction > 50% (%)	172 (50.1)	0 (0)	< 0.001
The average reduction of LDL-C (mg/dL)	−55.08 ± 36.86	−20.77 ± 36.78	< 0.001
The average percentage of LDL-C reduction (%)	−44.61 ± 25.12	−6.79 ± 32.86	< 0.001

Data are expressed as mean ± standard deviation or number (percentage). Abbreviation: LDL-C: low-density lipoprotein cholesterol.

**Table 5 jcm-09-00079-t005:** The univariate and multivariate logistic regression analyses of associations for non-achievement of LDL-C targets.

Variables	Univariate Analyses			Multivariate Analysis		
	OR	95% CI	*p*-value	OR	95% CI	*p*-value
Male	0.812	0.404–1.634	0.560			
Age	1.009	0.995–1.024	0.211			
BMI	1.006	0.980–1.032	0.671			
Diabetes mellitus	0.748	0.506–1.106	0.145	0.651	0.435–0.991	0.045
Current smoker	1.068	0.734–1.553	0.731			
Hypertension	1.284	0.880–1.874	0.195	1.461	0.975–2.985	0.066
Prior MI	1.402	0.587–3.347	0.447			
Prior stroke	0.693	0.268–1.791	0.449			
CKD stage ≥ 3 (eGFR < 60 mL/min/1.73 m^2^)	0.871	0.497–1.528	0.631			
Dyslipidemia with prior statin use	1.069	0.738–1.547	0.725			
Blood fasting sugar (mg/dL)	1.001	0.999–1.003	0.420			
HbA1C (%)	1.074	0.963–1.199	0.201	1.049	0.910–1.209	0.510
Total cholesterol (mg/dL)	0.997	0.993–1.001	0.165	0.999	0.993–1.004	0.620
LDL-cholesterol (mg/dL)	0.997	0.993–1.002	0.248			
Initial LDL-C >130 mg/dL	1.473	0.983–2.207	0.061	1.583	1.038–2.415	0.033
HDL-cholesterol (mg/dL)	1.001	0.984–1.017	0.943			
Initial HDL-C < 40 mg/dL	0.854	0.592–1.233	0.401			
Multiple vessel coronary disease	1.023	0.829–1.262	0.831			
Left main coronary artery disease	2.242	1.103–4.554	0.026	2.376	1.149–4.915	0.020
One-year post-MI angina	1.070	0.566–2.022	0.835			
One-year TVR	1.514	0.517–4.433	0.450			
One-year recurrent MI	0.795	0.153–4.142	0.786			

Abbreviation: OR: odds ratio; CI: confidence interval; LDL-C: low density lipoprotein cholesterol; BMI: body mass index; MI: myocardial infarction; CKD: chronic kidney disease; HbA1C: glycohemoglobin; HDL-C: high density lipoprotein cholesterol; TVR: target vessel revascularization.

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
