# Peer review of "The Clinical Outcomes Based on the Achievement of Low-Density Lipoprotein Cholesterol Targets after ST Elevation Myocardial Infarction"

_jcm, 2019, doi:10.3390/jcm9010079_

Round 1

Reviewer 1 Report

The study is limited with a single ethnic group but the authors are clear as to why the study design with moderate intensity statin was chosen.

please check the grammar and sentence building carefully as there are multiple places where changes need to be made.

Author Response

Specific responses to the first reviewer’s comments:

Reviewer #1:

Comment 1: The study is limited with a single ethnic group but the authors are clear as to why the study design with moderate intensity statin was chosen.

Responses: Thanks for your comments for our work.

Comment 2: Please check the grammar and sentence building carefully as there are multiple places where changes need to be made.

Responses: We sent the revised manuscript for English editing (MDPI's English editing) and have revised our manuscript. We provided the certificate of English editing.

Thank you for your constructive and valuable comments.

Reviewer 2 Report

Lee et al. described a study which demonstrated the comparison of clinical outcomes in STEMI patients with or without LDL-C target achievement. The cut-off values of LDL-C 70mg/dl and 50% reduction were used for the prediction of adverse clinical events. Although the findings are of potential interest, several major limitations and concerns should be addressed by the authors in the analysis.

Approximately 20% of the patients were excluded from the study due to various reasons. Were there any differences between patients with and without complete follow-up? The serial change of lipid profile in the 2 groups could be of interest. (The authors noted that the data was evaluated every 3-6months in the methods section.) The information about the change of medical therapy is needed (Increasement of the statin dose, adding ezetimibe, PCSK9 inhibitors, etc.). The authors have defined MACE by using their own criteria. To be consistent with the previous studies, the definition of MACE should be death or cardiac death, myocardial infarction, stroke (+ urgent revascularization). Primary endpoint was noted as ‘follow-up LDL-C level < 70 mg/dL and or ≥50% reduction compared to initial LDL-C level’. However, the aim of was noted as ‘the present study aimed to explore the clinical outcome in STEMI patients with or without achievement LDL-C target (below 70 mg/dL and/or ≥50% reduction) one year later after STEMI’. To be consistent with the aim, the clinical outcomes should be considered as a primary endpoint. In the statistic section, the authors mentioned that Mann-Whitney U test, which is a method for variables with skewed distribution, was used for the analysis. However, there is no variable expressed by median and inter-quartile values in the tables. Which variable required this method? Since the authors stated that the LDL-C levels was from patients under moderate-intensity statin therapy, the patients who discontinued statin therapy should be excluded. Any information about the QCA at follow-up? This could provide us more detailed information in terms of plaque regression/progression. Are there any differences between patients with and without prior statin therapy? Could the lipid-lowering therapy impact better on patients who were naïve to statin therapy? The authors discussed about the importance of lipid-lowering therapy in patients with LMT disease. However, the discussion should be focusing on the factor which cause LDL-C continuously high despite of statin therapy. Why did LMT, non-DM influence the LDL-C levels after statin therapy? This should be elaborated in the discussion section. Clinical implication and future perspective should be added in the discussion section. Recent clinical trials have proven the benefit of LDL-C lowering therapy as a secondary prevention by enrolling larger populations and longer follow-up periods. This should be addressed in the discussion and limitation. (e.g. N Engl J Med. 2019 Nov 18. and N Engl J Med. 2018 Nov 29;379(22):2097-2107.)

Author Response

Specific responses to the second reviewer’s comments:

Reviewer #2:

Comment 1: Approximately 20% of the patients were excluded from the study due to various reasons. Were there any differences between patients with and without complete follow-up?

Responses: Thanks for your comment. In our study, the patients without complete follow-up lipid data was related to transfer to local hospital or local clinic or no statin use. Therefore, we did not have enough data for analyses. We are very sorry about we could not provide the differences between patients with and without complete follow-up.

Comment 2: The serial change of lipid profile in the 2 groups could be of interest. (The authors noted that the data was evaluated every 3-6months in the methods section.)

Responses: We provided the revised figure 2 about the serial change of lipid profile in the 2 groups. We also added one paragraph in the revised manuscript in results, at page 16, paragraph 2, line 1-5, and at page 17, paragraph 1, line 1-5, as “In group 1, the average LDL-C level decreased to 72.39 ± 25.36 mg/dL at 6-month follow-up period and showed significant difference when comparing with initial LDL-C level (Figure 2). At one-year follow-up period, the average LDL-C level decreased to 54.24 ± 15.18 mg/dL and presented significant difference when comparing with 6-month LDL-C level. In group 2, the average LDL-C level decreased to 104.79 ± 28.20 mg/dL at 6-month follow-up period and showed significant difference when comparing with initial LDL-C level (Figure 2). At one-year follow-up period, the average LDL-C level decreased to 92.73 ± 22.48 mg/dL and presented significant difference when comparing with 6-month LDL-C level”. We also added the figure legends about figure 2.

Comment 3: The information about the change of medical therapy is needed (Increasement of the statin dose, adding ezetimibe, PCSK9 inhibitors, etc.).

Responses: We provided the revised manuscript in results, at page 10, paragraph 1, line 11-13, as “The percentage of increasement of the statin dose (group 1 vs. group 2; 16.0 % vs. 30.8 %; p<0.001) and ezetimibe use (group 1 vs. group 2; 2.6 % vs. 12.8 %; p<0.001) was significant higher in group 2”. We also added associated data in Table 1.

Comment 4: The authors have defined MACE by using their own criteria. To be consistent with the previous studies, the definition of MACE should be death or cardiac death, myocardial infarction, stroke (+ urgent revascularization).

Responses: Thanks for your comment. We need to modify the definition about MACE because the patients with early death already were excluded in our study. We only provide the results about recurrent myocardial infarction and target vessel revascularization and stroke. We deleted the definition of MACE.  

Comment 5: Primary endpoint was noted as ‘follow-up LDL-C level < 70 mg/dL and or ≥50% reduction compared to initial LDL-C level’. However, the aim of was noted as ‘the present study aimed to explore the clinical outcome in STEMI patients with or without achievement LDL-C target (below 70 mg/dL and/or ≥50% reduction) one year later after STEMI’. To be consistent with the aim, the clinical outcomes should be considered as a primary endpoint.

Responses: Thanks for your comment. We modified the definition about the primary endpoints. We modified the revised manuscript in study endpoints at page 8. Paragraph 2, line 1-5 as ”The primary endpoints were recurrent angina episode that needed admission or an emergency department visit, recurrent TVR, recurrent MI, and stroke during the one-year follow-up period. The secondary endpoints of our study were LDL-C level < 70 mg/dL and or ≥50% reduction compared to initial LDL-C level”.

Comment 6: In the statistic section, the authors mentioned that Mann-Whitney U test, which is a method for variables with skewed distribution, was used for the analysis. However, there is no variable expressed by median and inter-quartile values in the tables. Which variable required this method?

Responses: Thanks for your comment. We modified the revised manuscript in statistical analysis at page 8, paragraph 3, line 3-4 as “Continuous variables were compared using an independent t-test”.  

Comment 7: Since the authors stated that the LDL-C levels was from patients under moderate-intensity statin therapy, the patients who discontinued statin therapy should be excluded.

Responses: Thanks for your comment. We modified the revised manuscript in patients and groups at page 6, paragraph 3, line 3 as “A total of 126 patients without statin use and had no follow-up lipid profile …… were excluded”. We also modified Figure 1.

Comment 8: Any information about the QCA at follow-up? This could provide us more detailed information in terms of plaque regression/progression.

Responses: Thanks for your comment. We did not perform follow-up angiography for every patient, so we could not provide the information about QCA at follow-up. We are very sorry about this.

Comment 9: Are there any differences between patients with and without prior statin therapy? Could the lipid-lowering therapy impact better on patients who were naïve to statin therapy?

Responses: Thanks for your comment. In univariate logistic regression analyses, dyslipidemia and prior statin use was not a predictor about the achievement of LDL-C target. We added one paragraph in revised manuscript in discussion at page 21, paragraph 1, line 5-6, as ” Third, dyslipidemia and prior statin use was not a predictor about the achievement of LDL-C target”.

Comment 10: The authors discussed about the importance of lipid-lowering therapy in patients with LMT disease. However, the discussion should be focusing on the factor which cause LDL-C continuously high despite of statin therapy. Why did LMT, non-DM influence the LDL-C levels after statin therapy? This should be elaborated in the discussion section. Clinical implication and future perspective should be added in the discussion section.

Responses: Thanks for your comment. We added one paragraph in discussion at page 23, paragraph 1, line 1-3, as” Physicians may give more aggressive lipid control and sugar control in the DM patients. Therefore, non-DM became the predictor about non-achievement of LDL-C target”. We added one paragraph in discussion at page 23, paragraph 1, line 12-14, as” On the other hand, the patients with left main coronary artery disease had the possibilities about non-achievement of LDL-C target even though aggressive lipid-lower agents”.

Comment 11: Recent clinical trials have proven the benefit of LDL-C lowering therapy as a secondary prevention by enrolling larger populations and longer follow-up periods. This should be addressed in the discussion and limitation. (e.g. N Engl J Med. 2019 Nov 18. and N Engl J Med. 2018 Nov 29;379(22):2097-2107.)

Responses: Thanks for your comment. We added one paragraph in discussion at page 21, paragraph 2, line 9-13, as” In the recent study about proprotein convertase subtilisin–kexin type 9 (PCSK9) inhibitor use for secondary prevention after acute coronary syndrome, the patients with lower LDL-C level had better outcomes about recurrent ischemic cardiovascular events during the longer follow-up period”. We added one paragraph in limitation at page 23, paragraph 2, line 7-8, as” Another limitation was no PCSK-9 inhibitor was used during study period in our study”. We also added associated reference.

Thank you for your constructive and valuable comments.

Round 2

Reviewer 2 Report

 This is a comment to the response to the review comment 7. At the first version of the manuscript, the author noted that the patients who continued statin therapy at follow-up were 467 patients among 515 patients (Table 1). But in the new version, this was deleted and all patients were included to the analysis (515 patients were considered to have continuous statin therapy). This should be explained by the authors.   The author should explain why LMT could be 'an independent predictor' of non-achievement of LDL-C targets. The REASON should be provided since this is one of the main result of the study.  

Author Response

Specific responses to the reviewer’s comments:

Reviewer #2:

Comment 1: At the first version of the manuscript, the author noted that the patients who continued statin therapy at follow-up were 467 patients among 515 patients (Table 1). But in the new version, this was deleted and all patients were included to the analysis (515 patients were considered to have continuous statin therapy). This should be explained by the authors.

Responses: This was our mistake. 467 patients continued statin therapy with the same brand. In our study, 515 patients continued statin therapy. Among them, 108 patients increased the dosage of statin, and 31 patients were added ezetimibe, and 467 patients continued statin therapy with the same brand. Therefore, we corrected this mistake in table 1 and figure. We added the parameter about statin therapy with the same brand in Table 1.

Comment 2:The author should explain why LMT could be 'an independent predictor' of non-achievement of LDL-C targets. The REASON should be provided since this is one of the main result of the study.

Responses: We provided the revised manuscript and added one paragraph in discussion at page 10, paragraph 1, line 297-303, as “In the patients with left main coronary artery disease…… In one observational study about patients with the acute coronary syndrome, a higher prevalence of left main coronary artery disease was noted in patients who did not attain LDL target, even though the p-value did not reach a significant difference 25. In our study, the prevalence of initial LDL-C level ≥ 130 mg/dL was also higher in patients with left main coronary artery disease (non-LM vs. LM; 128/482 (26.6%) vs. 10/33 (30.3%); p=0.685). On the other hand, an initial LDL-C level ≥ 130 mg/dL was the major problem with non-achievement of LDL-C targets”. We also added the associated reference.

Thank you for your constructive and valuable comments.